# Evaluation of Different Reverse Osmosis Membranes for Textile Dyeing and Finishing Wastewater Reuse

**DOI:** 10.3390/membranes13040420

**Published:** 2023-04-08

**Authors:** Chunhai Wei, Yequan Lao, Rulu Ouyang, Guorui Zhang, Guijing Huang, Feilong Deng, Qicheng Tan, Genghao Lin, Hong Zhou

**Affiliations:** 1Department of Municipal Engineering, School of Civil Engineering, Guangzhou University, Guangzhou 510006, China; 17819188014@163.com (Y.L.); ouyangrulu2022@163.com (R.O.); m15802055586@163.com (G.H.); 18127263182@163.com (F.D.); 17725683654@163.com (Q.T.); m15017256643@163.com (G.L.); 2Key Laboratory for Water Quality and Conservation of the Pearl River Delta, Ministry of Education, Guangzhou 510006, China; 3Department of Civil and Environmental Engineering, Texas A&M University, College Station, TX 77843-3122, USA; guoruizhang9@gmail.com

**Keywords:** concentration polarization, orthogonal test, osmotic pressure, reverse osmosis, textile dyeing and finishing wastewater, water recovery ratio

## Abstract

Different commercial reverse osmosis (RO) membranes from Vontron and DuPont Filmtec were evaluated for textile dyeing and finishing wastewater (TDFW) reuse in China. All six tested RO membranes produced qualified permeate meeting TDFW reuse standards at a water recovery ratio (WRR) of 70% in single batch tests. The rapid decline of apparent specific flux at WRR over 50% was mainly ascribed to feed osmotic pressure increase caused by concentrating effects. Multiple batch tests using Vontron HOR and DuPont Filmtec BW RO membranes with comparable permeability and selectivity demonstrated the reproducibility and showed low fouling development. The occurrence of carbonate scaling on both RO membranes was identified by scanning electron microscopy and energy disperse spectroscopy. No obvious organic fouling was detected on both RO membranes by attenuated total reflectance Fourier transform infrared spectrometry. From the orthogonal tests, with an integrated RO membrane performance index (i.e., 25% rejection ratio of total organic carbon + 25% rejection ratio of conductivity + 50% flux ratio of final to initial) as a target, the optimal parameters were determined as WRR of 60%, cross-flow velocity (CFV) of 1.0 m/s, temperature (T) of 20 °C for both RO membranes, while trans-membrane pressures (TMP) of 2 and 4 MPa were optimal for Vontron HOR RO membrane and DuPont Filmtec BW RO membrane, respectively. Both RO membranes with the optimal parameters produced good permeate quality for TDFW reuse and kept a high flux ratio of final to initial, demonstrating the effectiveness of the orthogonal tests.

## 1. Introduction

Owing to high rejection performance (theoretically only water molecules pass through) [1], the reverse osmosis (RO) membrane has been widely used for water and wastewater treatment, such as seawater and brackish water desalination [2,3], pure water production [4], and high-quality wastewater reuse mainly from biologically treated wastewater [5,6,7]. Loeb and Sourirajan successfully prepared the first RO membrane using cellulose acetate with asymmetric structure, high salt rejection, and a high flux in 1960. DuPont developed a commercial hollow fiber RO membrane made from aromatic polyamide for application in seawater and brackish water desalination in the 1970s. Meanwhile, Dow Chemicals and Toyobo developed a commercial hollow fiber RO membrane made from cellulose triacetate. Filmtec developed a commercial flat-sheet thin film composite RO membrane using aromatic polyamide in the 1980s–1990s. In the past two decades, the selective permeability, anti-fouling, and oxidation tolerance of RO membranes were further enhanced, especially pure water permeability, which was increased by 20% via doping functional nano-materials (e.g., NaA molecular sieves and aquaporins) into RO membranes. A novel RO membrane with low cost, long life, high salt rejection, and high flux is still a key element for RO process development [8].

Driven by the rapid industrialization and urbanization in the past three decades, China has been a major RO market, especially for high-quality wastewater reuse within industrial sectors (e.g., thermal power, steel, coal/petro-chemicals, textile) [9,10,11,12,13]. Different from seawater desalination, industrial wastewater reuse via RO has been challenged by complex feed quality, high fouling potential, and short membrane life, demanding a high-performance RO membrane as well as high-efficiency process management [10,11,14,15,16,17].

Benefitting from early development, several RO membrane manufacturers including DuPont Filmtec, Hydranautics, Torary, Koch, Veolia Osmonics, and Toyobo are the leading international RO membrane suppliers [4]. Their mainstream RO membrane products are applied for seawater and brackish water desalination. Driven by the demand of industrial wastewater reuse (even zero liquid discharge) in China, some domestic RO membrane manufactures such as Vontron, Origin Water, and Ovay have begun to develop RO membranes targeting industrial wastewater reuse in recent years [13]. However, it is necessary to obtain intensive experience in various industrial wastewater applications for these domestic RO membrane products to be widely accepted by the end users.

The textile industry is one of the largest wastewater producers among all industrial sectors in China, discharging wastewater of 1.84 billion tons and accounting for 10.1% of the total industrial wastewater discharge amount in 2015 [18]. Its wastewater comes mainly from dyeing and finishing process, thus generally referred to as textile dyeing and finishing wastewater (TDFW). With a desalination-demanding TDFW reuse standard (FZ/T 01107-2011) effective since 2011, and a very stringent TDFW discharge standard (GB 4287-2012) effective since 2012 in China, TDFW reuse via RO has gradually been a double-win solution, which can reduce wastewater discharge as well as conserve freshwater consumption in the textile industry [10]. However, Chinese domestic RO membranes only have a minor market share for TDFW reuse, and thus require more operational experiences to advance their application.

In this study, RO membranes from a Chinese domestic manufacturer (Vontron, Guiyang, China) and a leading international manufacturer (DuPont Filmtec, Edina, MN, USA) were systematically evaluated for TDFW reuse in terms of inorganic and organic rejection, fouling development, and optimal process parameters. The outcome would provide technical supports for screening suitable RO membranes and their optimal operational parameters for TDFW reuse.

## 2. Materials and Methods

### 2.1. Lab-Scale RO Setup and RO Membranes

A constant-pressure cross-flow RO setup was used for the evaluation of different RO membranes in this study, and its detailed description is shown in a previous study [19]. Six kinds of flat-sheet polyamide-based thin-film composite RO membranes were selected for this study. Four RO membranes with model names of PURO (referring to anti-fouling improvement), LP (referring to low pressure operation), HOR (referring to high oxidation resistance), and SW (referring to seawater application) were supplied by Vontron. Two RO membranes with model names of BW and SW (referring to brackish water and seawater application, respectively) were supplied by DuPont Filmtec.

The specific pure water flux of new RO membranes was measured via pre-tests using pure water, under a temperature (T) of 20 °C, and a cross-flow velocity (CFV) of 1 m/s (shown in Figure 1). Pure membrane resistance (R_m_) was further calculated from Darcy’s Law (shown in Table 1). The rejection of Na^+^ and Mg^2+^ by new RO membranes was also monitored via pre-tests using a synthetic mixture of NaCl (2000 mg/L) and MgCl_2_ (2000 mg/L) under trans-membrane pressure (TMP) of 2 MPa, T of 20 °C, and CFV of 1 m/s. From Table 1, it is seen that BW from DuPont Filmtec had the highest permeability but lowest selectivity, PURO from Vontron had moderate permeability and the highest selectivity, both SWs from DuPont Filmtec and Vontron had similar lowest permeability and moderate selectivity, and LP and HOR from Vontron had similar moderate permeability and selectivity.

### 2.2. Wastewater Sample and Analytical Methods

Effluent from a membrane bioreactor (MBR) treating TDFW in Zhongshan, Guangdong Province, China, was sampled for RO feed. It looked clear and yellowish-brown without suspended solids (SS) due to the application of the ultrafiltration membrane with a mean pore size of 30 nm. Its quality and the TDFW reuse standard in China (FZ/T 01107-2011) are shown in Table 2. Water quality was measured according to the standard methods [20]. pH was measured with a pH meter (HQ4300, Hach, Ames, IA, USA). Chemical oxygen demand (COD) was measured via potassium dichromate oxidation followed by titration with ferrous ammonium sulfate. Total organic carbon (TOC) was measured with an organic carbon analyzer (TOC-L, Shimadzu, Kyoto, Japan). SS was measured via a heating and weighing method. Chroma was measured via a dilution visual colorimetric method. Fe, Mn, and total hardness as CaCO_3_ were measured with an inductively coupled plasma spectrometer (iCAP 7000, Thermo Scientific, Waltham, MA, USA). Conductivity was measured with a conductivity meter (HQ4300, Hach). Osmotic pressure was measured with a freezing point osmometer (Advanced 3300, Advanced Instruments, Norwood, MA, USA).

### 2.3. Batch RO Tests and Analytical Methods

The evaluation of different RO membranes for MBR effluent filtration was conducted in 4 phases. In Phase 1, single batch test using each RO membrane was conducted to compare its rejection performance and permeability decline with water recovery ratio (WRR) for preliminary screening candidate RO membranes. Based on new membrane permeability in Table 1, TMP was selected as 1.5, 2, 1, 1, 1.5, and 2 MPa for BW, SW from DuPont Filmtec, PURO, LP, HOR, and SW from Vontron, respectively, to achieve an initial flux of 20–30 L/(m^2^·h), a typical value in real RO application. CFV, T, and initial feed volume were 1 m/s, 20 °C, and 2 L, respectively, for all RO membranes. RO membrane permeate was sampled under different WRR (i.e., a ratio of permeate volume to initial feed volume) for analysis. RO membrane flux was continuously monitored for total resistance (R_t_) analysis. After MBR effluent filtration, specific pure water flux was measured again for each RO membrane to calculate pure membrane plus fouling resistance (R_m_ + R_f_). Resistance caused by concentration polarization (R_p_) was further calculated as R_t_ − R_m_ − R_f_. It is worth pointing out that the pressure in Darcy’s Law for RO membrane filtration should be the net value of applied pressure minus the osmotic pressure of RO feed.

In Phase 2, multiple batch tests using candidate RO membranes and the same methodology as in Phase 1 were further conducted to investigate the reproducibility and fouling development. After filtration, the used RO membranes were analyzed with a scanning electron microscope with an energy disperse spectroscope (SEM-EDS, Tescan Mira LMS, Kohoutovice, Czech) and a Fourier transform infrared spectrometer with attenuated total reflectance (ATR-FTIR, Thermo Scientific Nicolet iS20) for both inorganic scaling and organic membrane fouling characterization [21,22].

In Phase 3, orthogonal tests based on single batch test using candidate RO membranes were conducted to explore the optimal process parameters including T (20 °C, 30 °C, 40 °C), TMP (2, 3, 4 MPa), CFV (0.5, 1, 1.5 m/s), and WRR (60%, 70%, 80%), considering this methodology has been successfully applied in RO studies [23,24,25]. Thus, a L_9_ orthogonal table with a factor number of 4 and level number of 3 (i.e., 3^4^) was generated with a total batch test number of 9. T, TMP, CFV, and WRR were adjusted to the target value via chiller, valve, pump flow, and permeate weight, respectively. The TOC rejection ratio (i.e., TOC_r_), conductivity rejection ratio (i.e., Cond_r_), and the ratio of final flux to initial flux (i.e., J_f_/J_i_) were selected as the RO membrane performance indices evaluating the selectivity and permeability of RO membranes. For evaluating the whole performance of RO membranes via orthogonal tests, a single performance index (i.e., P) was proposed to integrate the selectivity and permeability of RO membranes with different weight coefficients in this study, which was defined as P = 0.25TOC_r_ + 0.25Cond_r_ + 0.5J_f_/J_i_.

In Phase 4, single batch test using candidate RO membranes with the optimal process parameters from Phase 3 was conducted to demonstrate the outcome of orthogonal tests because the combination of optimal process parameters was generally not conducted in orthogonal tests.

## 3. Results and Discussion

### 3.1. Candidate RO Membranes Selection via Single Batch Test for TDFW reuse in Phase 1

As shown in Figure 2, both permeate conductivity and TOC increased with water recovery ratio for all RO membranes. This was attributed to the concentrating effects from RO membranes. Both inorganic and organic substances were efficiently rejected by RO membranes and thus concentrated in feed with water recovery ratio, resulting in the elevated concentration in permeate. Benefitting from the high selectivity of all RO membranes used in this study, all permeate quality at a water recovery ratio of as high as 70% could meet the TDFW reuse standard in China (shown in Table 3).

As shown in Figure 3a, the apparent specific flux showed an accelerated decline with water recovery ratio for all RO membranes, especially after a water recovery ratio of 50%. However, the flux decline in the constant-pressure RO filtration test was ascribed to both resistance increase caused by concentration polarization and membrane fouling, and true TMP decrease from feed osmotic pressure increase caused by concentrating effects. In this study, feed osmotic pressure was measured at water recovery ratios of 10%, 30%, 50%, and 70%, respectively. After subtracting feed osmotic pressure, the corrected specific flux was significantly higher than the apparent one, especially at water recovery ratios of 50% and 70%, indicating the dominant contribution to flux decline from true TMP decrease. From Figure 3b, it is seen that membrane fouling resistance R_m_ contributed 15–19% of the total resistance for all RO membranes, indicating less fouling occurrence which was mainly caused by only one batch filtration. Pure membrane resistance R_m_ was the major component accounting for 61–74% of total resistance for all other RO membranes except for the DuPont Filmtec BW RO membrane. Concentration polarization resistance R_p_ showed a large variation from 6% of the total resistance for Vontron PURO and LP RO membranes, to 12–13% for both SW RO membranes, to 20% for the Vontron HOR RO membrane, and to 45% for the DuPont Filmtec BW RO membrane, which might be related to the specific surface property (e.g., roughness) of each RO membrane [26].

Considering that all RO membranes tested in this study could produce qualified permeate for TDFW reuse, RO membrane screening was judged from permeability. Based on the comparable specific flux and its decline ratio, Vontron HOR and DuPont Filmtec BW were selected as candidate RO membranes for the following tests.

### 3.2. Reproducibility and Fouling Development of Candidate RO Membranes in Phase 2

As shown in Figure 4, the apparent specific flux showed a similar accelerated decline with water recovery ratio in three consecutive batches, and a gradual decrease with batch number for both candidate RO membranes, indicating a significant fouling development especially for the Vontron HOR RO membrane. 

As shown in Figure 5, some particles including big prisms and small spheres were found on both fouled RO membranes. Further elemental composition analysis via EDS (shown in Table 4) showed that C, O, Ca, Fe, Al, and Si were the major elements of these particles, implying the occurrence of carbonate scaling. Inorganic scaling, such as of carbonate and sulfate, was the common fouling phenomena on the RO membrane for seawater desalination and wastewater reuse as well [27,28].

As shown in Figure 6, both new membranes showed similar characteristic peaks of polyamides, indicating their similar active layer compositions, which were also comparable with previous studies [16,29]. However, there were no obvious changes in ATR-FTIR spectra between fouled and new membranes, implying no occurrence of significant organic fouling in the three batch filtrations, which was also in agreement with the above-mentioned SEM-EDS analysis.

### 3.3. Parameters Optimization for Candidate RO Membranes via Orthogonal Tests in Phase 3

As shown in Table 5, for Vontron HOR and DuPont Filmtec BW RO membranes under the same 9 orthogonal tests, TOC_r_, Cond_r_, and J_f_/J_i_ were in the ranges of 79.94–99.85% and 74.95–97.69%, 79.86–99.54% and 76.78–97.37%, and 45.29–82.17% and 49.56–77.50%, respectively. Both RO membranes showed comparable performances for TDFW reuse applications.

As shown in Table 6, the range analysis of orthogonal tests for Vontron HOR RO membranes showed that WRR and CFV were the major parameters affecting TOC_r_, Cond_r_, and J_f_/J_i_. For the integrated RO membrane performance index P, WRR had the greatest impact, followed by CFV, T, and TMP in descending sequence. The optimal parameters based on the orthogonal tests in this study were determined as a WRR of 60%, CFV of 1.0 m/s, T of 20 °C, and TMP of 2 MPa.

As shown in Table 7, a range analysis of orthogonal tests for DuPont Filmtec BW RO membranes showed that T and WRR were the major parameters affecting TOC_r_ and Cond_r_, while WRR and TMP were the major parameters affecting J_f_/J_i_. WRR, T, TMP, and CFV was in descending sequence affecting P. The optimal parameters based on the orthogonal tests in this study were determined as a WRR of 60%, T of 20 °C, TMP of 4 MPa, and CFV of 1.0 m/s.

### 3.4. Performance of Candidate RO Membranes with the Optimal Parameters in Phase 4

Based on the orthogonal tests, the demonstration tests were conducted for both candidate RO membranes with the optimal parameters. As shown in Table 8 for the Vontron HOR RO membrane, the demonstration test (test no. 10) ranked the top 3 major permeate qualities (e.g., COD, TOC, conductivity) and top 1 J_f_/J_i_ among the orthogonal tests, demonstrating the effectiveness of the methodology used in this study. As shown in Table 9 for the DuPont Filmtec BW RO membrane, the demonstration test (test no. 10) ranked the top 2 major permeate qualities (e.g., COD, TOC, conductivity) and top 1 J_f_/J_i_ among the orthogonal tests, also demonstrating the effectiveness of the methodology used in this study. Both RO membranes with the optimal parameters showed comparable performance, including similar qualified permeate for TDFW reuse and nearly the same flux decline during filtration.

## 4. Conclusions

All six RO membranes from Vontron and DuPont Filmtec produced qualified permeate, meeting TDFW reuse standards in China with a water recovery ratio of 70% in single batch filtration tests. Feed osmotic pressure increase caused by concentrating effects was the important factor affecting flux decline. Based on similar permeability and selectivity, Vontron HOR and DuPont Filmtec BW RO membranes were selected for multiple batch filtration tests, where the reproducibility was confirmed and fouling development was monitored. SEM-EDS analysis showed the occurrence of carbonate scaling on both RO membranes. ATR-FTIR analysis showed little organic fouling development in only three batch filtrations. From the orthogonal tests, the optimal parameters were determined as being a WRR of 60%, CFV of 1.0 m/s, T of 20 °C for both RO membranes, while TMP values of 2 and 4 MPa were optimal for Vontron HOR and DuPont Filmtec BW RO membranes, respectively. The impact order was WRR > CFV > T ≥ TMP for the Vontron HOR RO membrane, while it was WRR > T > TMP > CFV for the DuPont Filmtec BW RO membrane. The differences in membrane surface characteristics might be the reason behind this. Both RO membranes with optimal parameters from the orthogonal tests produced good quality permeate for TDFW reuse, and kept high J_f_/J_i_, demonstrating the effectiveness of the orthogonal tests.

## Figures and Tables

**Figure 1 membranes-13-00420-f001:**
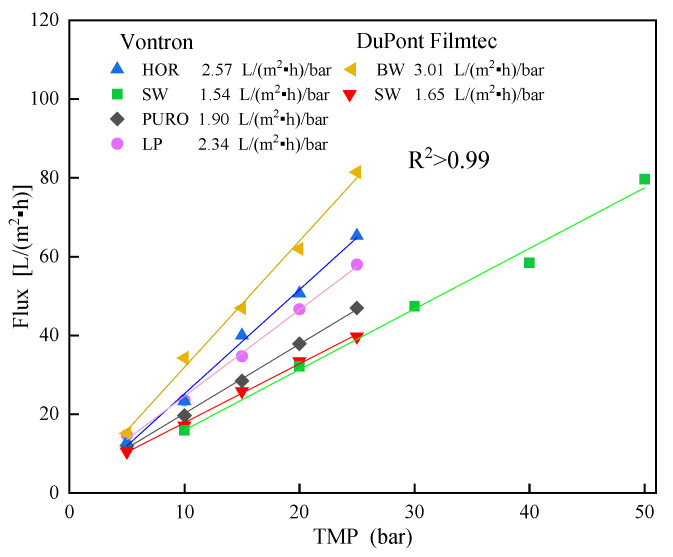
Changes in pure water flux with TMP for new RO membranes.

**Figure 2 membranes-13-00420-f002:**
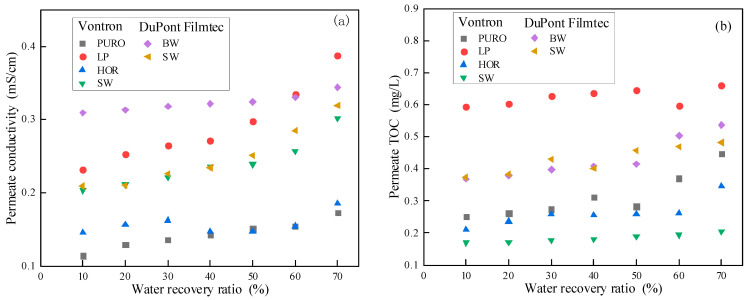
Changes in permeate conductivity (**a**) and TOC (**b**) of each RO membrane with water recovery ratio.

**Figure 3 membranes-13-00420-f003:**
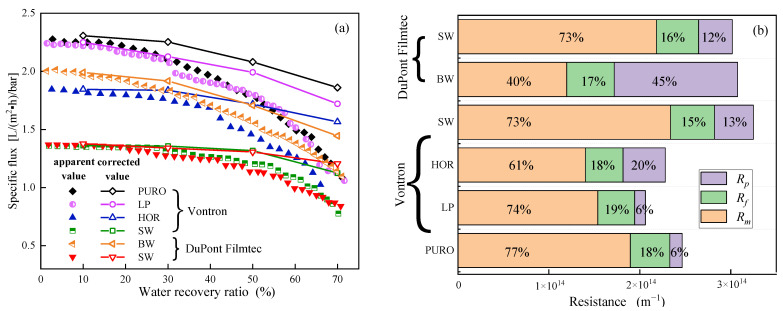
Changes in apparent and corrected specific flux with water recovery ratio (**a**) and resistance composition after filtration (**b**) for each RO membrane.

**Figure 4 membranes-13-00420-f004:**
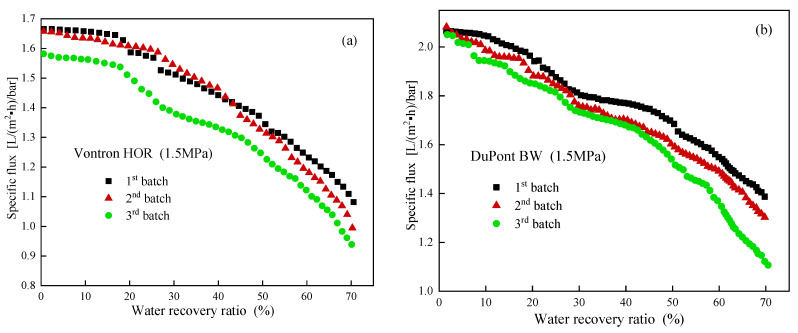
Changes in apparent specific flux with water recovery ratio for Vontron HOR (**a**) and DuPont Filmtec BW (**b**) RO membrane in 3 batch filtrations.

**Figure 5 membranes-13-00420-f005:**
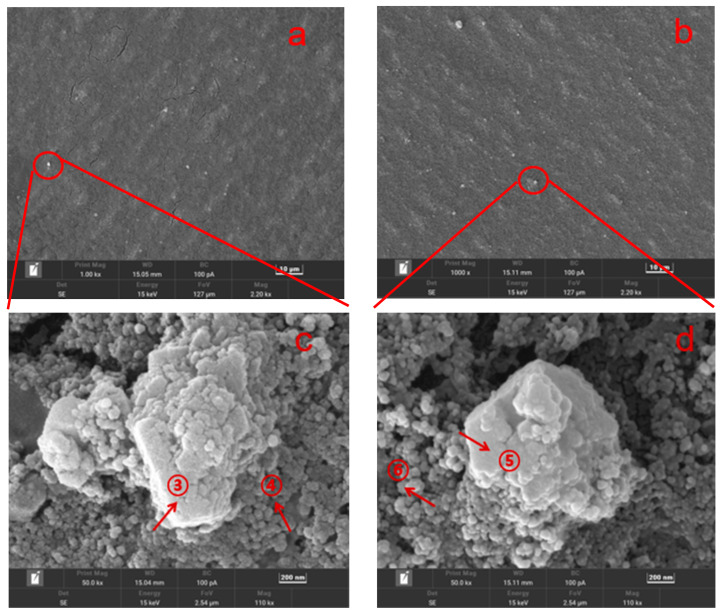
SEM images of fouled Vontron HOR ((**a**), 1000×; (**c**), 50,000×) and DuPont Filmtec BW ((**b**), 1000×; (**d**), 50,000×) RO membrane after 3 batch filtrations.

**Figure 6 membranes-13-00420-f006:**
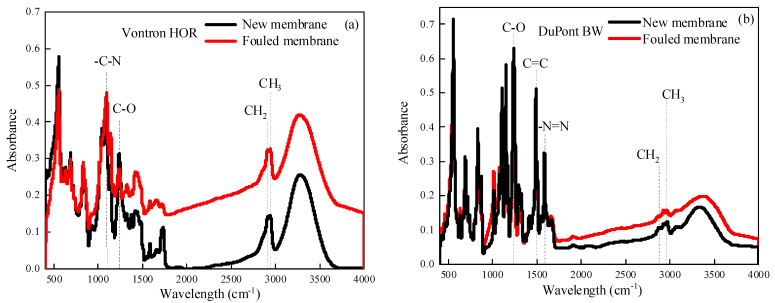
ATR-FTIR spectra of new and fouled Vontron HOR (**a**) and DuPont Filmtec BW (**b**) RO membrane after 3 batch filtrations.

**Table 1 membranes-13-00420-t001:** Permeability and selectivity of new RO membranes.

Manufacturer	Model Name	Specific Pure Water Flux at 20 °C [L/(m^2^·h)/bar]	Pure Membrane Resistance (×10^14^ m^−1^)	Na^+^ Rejection Ratio (%)	Mg^2+^ Rejection Ratio (%)
Vontron	PURO	1.90	1.90	99.18	99.81
	LP	2.34	1.54	98.51	99.71
	HOR	2.57	1.40	98.98	99.76
	SW	1.54	2.34	98.89	99.76
DuPont Filmtec	BW	3.01	1.20	96.30	98.46
	SW	1.65	2.18	98.75	99.34

**Table 2 membranes-13-00420-t002:** MBR effluent quality and the TFDW reuse standard in China.

Water Type	pH	COD (mg/L)	TOC (mg/L)	SS (mg/L)	Chroma (Dilution Times)	Fe (mg/L)	Mn (mg/L)	Total Hardness as CaCO_3_ (mg/L)	Conductivity (ms/cm)
MBR effluent	8.84	71.6	14.3	0	90	6	0.4	310.8	5.22
Reuse water	6.5–8.5	≤50	-	≤30	≤25	≤0.3	≤0.2	≤450	≤2.5

**Table 3 membranes-13-00420-t003:** Permeate quality of each RO membrane at water recovery ratio of 70%.

Manufacturer	Model Name	pH	COD (mg/L)	TOC (mg/L)	SS (mg/L)	Chroma (Dilution Times)	Fe (mg/L)	Mn (mg/L)	Total Hardness as CaCO_3_ (mg/L)	Conductivity (mS/cm)
Vontron	PURO	7.20	2.5	0.37	0	1	0.019	0.003	2.45	0.173
	LP	7.88	4.6	0.60	0	1	0.020	0.002	4.27	0.388
	HOR	7.98	1.9	0.26	0	1	0.029	0.003	2.96	0.185
	SW	7.95	1.5	0.19	0	1	0.018	0.005	3.35	0.202
DuPont Filmtec	BW	8.04	3.3	0.50	0	1	0.082	0.006	5.55	0.344
	SW	7.99	3.1	0.47	0	1	0.088	0.006	3.17	0.319

**Table 4 membranes-13-00420-t004:** Elemental composition of foulants on candidate RO membranes via EDS.

Vontron HOR RO Membrane						DuPont Filmtec BW RO Membrane					
Point Scanning ③		Point Scanning ④		Aera Scanning		Point Scanning ⑤		Point Scanning ⑥		Aera Scanning	
Element	wt%	Element	wt%	Element	wt%	Element	wt%	Element	wt%	Element	wt%
O	43.14	C	43.60	C	49.40	O	41.43	C	65.72	C	58.32
C	26.95	O	24.54	O	24.8	C	26.91	O	20.41	O	21.78
Ca	16.80	Fe	12.73	Fe	6.99	Ca	22.86	Fe	3.97	Ca	4.44
Fe	3.80	Al	5.74	Ca	5.30	Fe	2.03	Na	2.51	Fe	3.75
Al	3.02	Si	5.52	Al	4.30	Na	1.96	Cl	1.84	Na	2.61
Si	2.64	Na	3.68	Si	3.93	N	1.16	Al	1.75	Cl	2.29
Na	2.17	Ca	1.80	Na	2.64	Al	1.20	Si	1.53	Al	2.04
Cl	0.41	Cl	0.98	Cl	0.81	Si	0.95	Mn	0.76	N	1.87
Mg	0.33	N	0.55	N	0.67	Cl	0.84	Ca	0.64	Si	1.74
Mn	0.30	K	0.36	S	0.63	Mg	0.34	K	0.48	Mn	0.37
Zn	0.27	Cu	0.25	K	0.22	Mn	0.28	Mg	0.39	S	0.35
K	0.19	Mg	0.21	Mg	0.21	K	0.04			K	0.22
		Zn	0.04	Mn	0.05					Mg	0.12
				Zn	0.05					Cu	0.08

**Table 5 membranes-13-00420-t005:** Results of the orthogonal tests for candidate RO membranes.

Test No.	Operational Parameters				Operational Performance of Vontron HOR RO Membrane			Operational Performance of DuPont Filmtec BW RO Membrane		
	TMP (MPa)	CFV (m/s)	T (°C)	WRR (%)	TOC_r_ (%)	Cond_r_ (%)	J_f_/J_i_ (%)	TOC_r_ (%)	Cond_r_ (%)	J_f_/J_i_ (%)
1	2	0.5	20	60	81.95	82.93	78.86	97.69	97.37	70.88
2	2	1	25	70	99.85	99.32	71.93	79.27	80.56	62.61
3	2	1.5	30	80	99.23	98.86	45.29	77.49	77.69	49.56
4	3	0.5	25	80	79.94	79.86	53.37	74.95	76.78	55.71
5	3	1	30	60	86.15	87.20	82.17	78.48	80.14	77.50
6	3	1.5	20	70	99.83	99.54	70.17	94.22	93.60	66.84
7	4	0.5	30	70	89.14	87.79	67.13	85.04	82.78	71.90
8	4	1	20	80	87.11	85.93	59.70	89.34	80.98	70.66
9	4	1.5	25	60	85.79	86.82	81.46	88.07	87.56	71.71

**Table 6 membranes-13-00420-t006:** Range analysis of the orthogonal tests for Vontron HOR RO membrane.

Performance Index		Operational Parameters				Summary
		TMP	CFV	T	WRR	
TOC_r_ (%)	A_1_	93.68	83.68	89.63	84.63	Quantitative impact order: WRR ≥ CFV > TMP > T Optimal parameters: WRR 70%, CFV 1.5 m/s, TMP 2 MPa, T 30 °C
	A_2_	88.64	91.03	88.53	96.27	
	A_3_	87.34	94.95	91.51	88.76	
	R	6.33	11.28	2.98	11.64	
Cond_r_ (%)	A_1_	93.70	83.52	89.47	85.65	Quantitative impact order: CFV > WRR > TMP > T Optimal parameters: CFV 1.5 m/s, WRR 70%, TMP 2 MPa, T 30 °C
	A_2_	88.87	90.82	88.67	95.55	
	A_3_	86.84	95.07	91.28	88.22	
	R	6.86	11.55	2.62	9.90	
J_f_/J_i_ (%)	A_1_	65.36	66.46	69.58	80.83	Quantitative impact order: WRR >> CFV > T > TMP Optimal parameters: WRR 60%, CFV 1.0 m/s, T 20 °C, TMP 4 MPa
	A_2_	68.57	71.27	68.92	69.74	
	A_3_	69.43	65.64	64.86	52.79	
	R	4.07	5.62	4.71	28.04	
P (%)	A_1_	79.52	75.03	79.56	82.99	Quantitative impact order: WRR > CFV > T ≥ TMP Optimal parameters: WRR 60%, CFV 1.0 m/s, T 20 °C, TMP 2 MPa
	A_2_	78.66	81.10	78.76	82.83	
	A_3_	78.26	80.33	78.13	70.64	
	R	1.26	6.07	1.43	12.35	

**Table 7 membranes-13-00420-t007:** Range analysis of the orthogonal tests for DuPont Filmtec BW RO membrane.

Performance Index		Operational Parameters				Summary
		TMP	CFV	T	WRR	
TOC_r_ (%)	A_1_	84.82	85.89	93.75	88.08	Quantitative impact order: T > WRR > TMP > CFV Optimal parameters: T 20 °C, WRR 60%, TMP 4 MPa, CFV 1.5 m/s
	A_2_	82.55	82.37	80.76	86.17	
	A_3_	87.49	86.59	80.34	80.60	
	R	4.94	4.23	13.41	7.49	
Cond_r_ (%)	A_1_	85.21	85.64	90.65	88.36	Quantitative impact order: T ≥ WRR > CFV > TMP Optimal parameters: T 20 °C, WRR 60%, CFV 1.5 m/s, TMP 2 MPa
	A_2_	83.50	80.56	81.63	85.64	
	A_3_	83.77	86.28	80.20	78.48	
	R	1.70	5.73	10.45	9.88	
J_f_/J_i_ (%)	A_1_	62.07	66.16	69.46	73.36	Quantitative impact order: WRR > TMP > CFV > T Optimal parameters: WRR 60%, TMP 4 MPa, CFV 1.0 m/s, T 20 °C
	A_2_	66.68	71.31	64.40	68.17	
	A_3_	71.42	62.70	66.32	58.64	
	R	9.35	8.61	5.06	14.72	
P (%)	A_1_	73.54	75.97	80.83	80.79	Quantitative impact order: WRR > T > TMP > CFV Optimal parameters: WRR 60%, T 20 °C, TMP 4 MPa, CFV 1.0 m/s
	A_2_	74.85	76.39	72.80	77.04	
	A_3_	78.53	74.57	73.29	69.09	
	R	4.98	1.82	8.03	11.70	

**Table 8 membranes-13-00420-t008:** Permeate quality and final flux of Vontron HOR RO membrane in the orthogonal (No. 1–9) and demonstration (No. 10) tests.

Test No.	pH	COD (mg/L)	TOC (mg/L)	Fe (mg/L)	Mn (mg/L)	Total Hardness as CaCO_3_ (mg/L)	Conductivity (mS/cm)	J_f_/J_i_ (%)
1	8.46	25.6	3.65	0.089	0.029	2.37	2.510	78.9
2	7.93	0.2	0.03	0.017	0.005	1.86	0.085	71.9
3	7.79	0.8	0.12	0.007	0.003	1.93	0.125	45.3
4	8.33	21.3	3.04	0.058	0.015	2.38	2.220	53.4
5	8.25	19.6	2.80	0.028	0.011	2.17	1.881	82.2
6	7.65	0.2	0.03	0.009	0.003	1.94	0.058	70.2
7	8.14	13.2	1.88	0.021	0.012	2.04	1.539	67.1
8	8.2	13.7	1.96	0.032	0.013	2.25	1.551	59.7
9	8.21	20.1	2.87	0.036	0.004	2.42	1.938	81.5
10	7.64	0.7	0.05	0.012	0.005	1.94	0.079	82.7

**Table 9 membranes-13-00420-t009:** Permeate quality and final flux of DuPont Filmtec BW RO membrane in the orthogonal (No. 1–9) and demonstration (No. 10) tests.

Test No.	pH	COD (mg/L)	TOC (mg/L)	Fe (mg/L)	Mn (mg/L)	Total Hardness as CaCO_3_ (mg/L)	Conductivity (mS/cm)	J_f_/J_i_ (%)
1	7.73	3.3	0.47	0.042	0.007	1.69	0.386	80.3
2	8.41	25.2	3.59	0.05	0.017	2.60	2.450	65.8
3	8.56	23.9	3.42	0.05	0.016	2.73	2.460	49.6
4	8.41	26.6	3.80	0.055	0.022	2.91	2.560	55.7
5	8.34	30.5	4.35	0.057	0.016	2.47	2.920	77.5
6	8.08	7.0	1.05	0.023	0.006	2.16	0.807	66.8
7	8.35	18.2	2.59	0.037	0.012	2.36	2.170	67.0
8	8.34	11.3	1.62	0.047	0.014	2.04	2.097	64.9
9	8.05	16.9	2.41	0.046	0.007	2.13	1.828	71.7
10	8.10	6.9	0.84	0.055	0.006	1.94	0.704	82.5

## Data Availability

Not applicable.

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
