# Peer review of "Evaluation of Different Reverse Osmosis Membranes for Textile Dyeing and Finishing Wastewater Reuse"

_membranes, 2023, doi:10.3390/membranes13040420_

Round 1

Reviewer 1 Report

The manuscript by Wei et. al presented the removal of TOC and dyes from textile water. The recovery is above 70%. The study is in-depth and gives strong evidence regarding the performance of 6 different membranes. The result is persuasive that all the commercial membranes in China and US have the same capability of separating TOCs and dyes from wastewater. The paper is well-suitable for publication on Membranes.

The term"Specific flux" used in some figure labels are well known as"permeance", and its unit commonly used is LMH*bar-1. Please kindly check.

Author Response

Comment 1: The manuscript by Wei et. al presented the removal of TOC and dyes from textile water. The recovery is above 70%. The study is in-depth and gives strong evidence regarding the performance of 6 different membranes. The result is persuasive that all the commercial membranes in China and US have the same capability of separating TOCs and dyes from wastewater. The paper is well-suitable for publication on Membranes.

Response: Thanks for your approval on our manuscript.

Comment 2: The term"Specific flux" used in some figure labels are well known as"permeance", and its unit commonly used is LMH*bar-1. Please kindly check.

Response: Thanks for your comments. As the term characterizing membrane flux per unit trans-membrane pressure, "specific flux" has been widely used in the field of membrane technology for water and wastewater treatement. Another term with the same meaning as "specific flux", "permeability" has also been wildely used. We kept "specific flux" in the revised manuscript. As for the unit of specific flux, we changed into LMH/bar following your suggestions in the revised manuscript.

Reviewer 2 Report

Evaluation of Different Reverse Osmosis membranes...

The paper describes the research on several membranes designed for reverse osmosis (RO). Out of the six flat-sheet membranes tested, two were selected based on preliminary tests for their favorable characteristics, i.e., optimal properties for treating textile wastewater.
Membrane processes require a range of desirable features such as high selectivity, high permeability, low fouling, high rejection ratio, and final to initial flux ratio. In practice, these characteristics are interdependent, and an optimally selected membrane for a specific process must reconcile these conflicting trends. In such cases, multiobjective optimization is used. The authors subjectively assigned weights to the desired parameters and used these criteria to select two membranes for further study. They then chose the "orthogonal test" methodology to find the optimal operating parameters for these membranes.
The paper lacks a reference to the orthogonal test method, which would demonstrate the mathematical or statistical validity of this experimental design method.
The paper is written in understandable language but contains minor grammatical errors.

It is debatable to use "orthogonal test" methodology while "Surface response methodology" seems better for finding optimal conditions for RO processes. The parameters found and described in lines 235-236 and Tables 6 and 7 as "The optimal parameters were determined as WRR of 60%, CFV of 1.0 m/s, T of 20°C, and TMP of 2 MPa" are not mathematically speaking optimal but rather suboptimal since each of them was chosen from three tested parameters. Thus, the cited sentence should be formulated as: "The near (or sub) optimal parameters or the “best results” were determined for WRR of 60%, CFV of 1.0 m/s, T of 20°C, and TMP of 2 MPa."
The TOCr and Condr terms used in Tables 5, 6, and 7 are not clearly defined in the paper.
The authors use the word "impaction" in the tables - it seems that they should use "impact order".
Minor editorial changes:
Line 137 - "It was worthy pointing out that" should be changed to "It is worthy pointing out that."
Line 143 - "a fourier transform infrared spectrometer..." should be changed to "Fourier transform infrared spectrometer..."
Line 182-183 - and further in Fig. 3 - the authors use the term "true specific flux" for specific flux corrected for osmotic pressure. However, "apparent specific flux" is true because it is measured. I suggest naming these terms "specific flux" and "corrected specific flux."
Table 9 - Conductivity (ms/cm) - should be changed to (mS/cm).

If the authors agree with these suggestions, please make the changes in the Abstract and Conclusions.

Author Response

Comment 1:The paper describes the research on several membranes designed for reverse osmosis (RO). Out of the six flat-sheet membranes tested, two were selected based on preliminary tests for their favorable characteristics, i.e., optimal properties for treating textile wastewater.

Membrane processes require a range of desirable features such as high selectivity, high permeability, low fouling, high rejection ratio, and final to initial flux ratio. In practice, these characteristics are interdependent, and an optimally selected membrane for a specific process must reconcile these conflicting trends. In such cases, multiobjective optimization is used. The authors subjectively assigned weights to the desired parameters and used these criteria to select two membranes for further study. They then chose the "orthogonal test" methodology to find the optimal operating parameters for these membranes.

The paper lacks a reference to the orthogonal test method, which would demonstrate the mathematical or statistical validity of this experimental design method.

The paper is written in understandable language but contains minor grammatical errors.

Response: Thanks for your approval on our manuscript. We added several references on the orthogonal test method and corrected the grammatical errors following your suggestions in the revised manuscript.

Comment 2:It is debatable to use "orthogonal test" methodology while "Surface response methodology" seems better for finding optimal conditions for RO processes. The parameters found and described in lines 235-236 and Tables 6 and 7 as "The optimal parameters were determined as WRR of 60%, CFV of 1.0 m/s, T of 20°C, and TMP of 2 MPa" are not mathematically speaking optimal but rather suboptimal since each of them was chosen from three tested parameters. Thus, the cited sentence should be formulated as: "The near (or sub) optimal parameters or the “best results” were determined for WRR of 60%, CFV of 1.0 m/s, T of 20°C, and TMP of 2 MPa."

The TOCr and Condr terms used in Tables 5, 6, and 7 are not clearly defined in the paper.

The authors use the word "impaction" in the tables - it seems that they should use "impact order".

Response: Thanks for your comments. We agree with your opinion about the orthogonal test methodology and response surface methodology. Theoretically response surface methodology should be better. As a classical methodology, orthogonal test has advantages of saving test number and reliable results. We kept the term “optimal parameters” but added the determiner “based on the orthogonal tests in this study” in the revised manuscript. The TOCr and Condr terms has been defined in Line 151-153 in Section 2.3. We replaced “impaction” with “impact order” in the revised manuscript.

Comment 3:Minor editorial changes:

Line 137 - "It was worthy pointing out that" should be changed to "It is worthy pointing out that."

Line 143 - "a fourier transform infrared spectrometer..." should be changed to "Fourier transform infrared spectrometer..."

Line 182-183 - and further in Fig. 3 - the authors use the term "true specific flux" for specific flux corrected for osmotic pressure. However, "apparent specific flux" is true because it is measured. I suggest naming these terms "specific flux" and "corrected specific flux."

Table 9 - Conductivity (ms/cm) - should be changed to (mS/cm).

Response: Thanks for your comments. We made the corresponding changes following your suggestions in the revised manuscript.

Comment 4:If the authors agree with these suggestions, please make the changes in the Abstract and Conclusions.

Response: Thanks for your comment. We made the above-mentioned changes in the Abstract and Conclusions in the revised manuscript.